# Research on Genotoxicity Evaluation of the Fungal Alpha-Amylase Enzyme on *Drosophila melanogaster*

**DOI:** 10.3390/biology14030219

**Published:** 2025-02-20

**Authors:** Arzu Taşpınar Ünal, Fahriye Zemheri Navruz, Safiye Elif Korcan, Sinan İnce, Emine Uygur Göçer

**Affiliations:** 1Department of Agricultural Biotechnology, Faculty of Agriculture, Iğdır University, Iğdır 76000, Türkiye; emine.uygur@igdir.edu.tr; 2Department of Molecular Biology and Genetics, Faculty of Science, Bartın University, Bartın 74110, Türkiye; fahriyezmhr@hotmail.com; 3Vocational School of Health Services, Uşak University, Uşak 64100, Türkiye; elif.korcan@usak.edu.tr; 4Department of Pharmacology and Toxicology, Faculty of Veterinary Medicine, Afyon Kocatepe University, Afyonkarahisar 03200, Türkiye; incesinan@gmail.com

**Keywords:** thermostable alpha-amylase, *Drosophila melanogaster*, genotoxicity

## Abstract

This study contributes to the biosafety assessment of microbial alpha-amylase enzymes widely used in industry. The developmental, genotoxic, and oxidative stress effects of alpha-amylase derived from Aspergillus niger G2-1 were examined in Drosophila melanogaster. The results indicate that low-dose alpha-amylase is safe, whereas high doses reduce larval survival rates and cause DNA damage. This study highlights the need for biosafety evaluation of enzyme exposure and provides new insights into this field.

## 1. Introduction

A native alpha-amylase enzyme produced by *Aspergillus niger* G2-1 does not cause oxidative stress and genotoxicity in the model organism *Drosophila melanogaster* within the recorded distribution and exposure time.

The hypothesis of this study is that exposure to different concentrations of thermostable alpha-amylase enzyme (produced from Aspergillus niger G2-1 isolate) may induce developmental, genotoxic, and oxidative stress effects in *Drosophila melanogaster*. Specifically, it is predicted that low-dose alpha-amylase is safe, whereas high doses (≥50 mg/mL) will reduce larval survival rates, cause DNA damage in adult flies, and potentially alter oxidative stress parameters.

Thermostable α-amylase is an enzyme widely used in industry. It has an important role, especially in the food, textile, biofuel, pharmaceutical, and paper industries, due to its ability to maintain its activity at high temperatures. Therefore, determining the genotoxic potential of this enzyme, which can be obtained from different sources, is critical for both human health and environmental safety.

Recent studies have evaluated the safety of α-amylase enzymes produced by various microorganisms. For instance, the European Food Safety Authority (EFSA) assessed an α-amylase produced by a non-genetically modified *Aspergillus niger* strain DP-Azb60. The evaluation included genotoxicity tests and a 90-day oral toxicity study in rats, which identified a no-observed-adverse-effect level (NOAEL) at the highest dose tested. The study concluded that this α-amylase does not raise safety concerns under its intended conditions of use. Similarly, EFSA evaluated an α-amylase from a genetically modified *Aspergillus niger* strain NZYM-MC. The assessment encompassed genotoxicity tests and a 90-day oral toxicity study in rodents, establishing a NOAEL at the highest dose tested. The findings indicated no safety concerns for this enzyme under its intended use conditions. These evaluations suggest that α-amylase enzymes, when produced and used under specified conditions, do not pose significant toxicity risks. Studies directly targeting the genotoxic effects of α-amylase are very limited. More comprehensive and specific studies in this area will contribute to a better understanding of the genotoxic potential of α-amylase.

This study is important because it shows for the first time that alpha-amylase obtained from the non-genetically modified local isolate *Aspergillus niger* G2-1 has no toxic effects and can be used safely in industry.

The alpha-amylase enzyme was produced on an industrial scale for the first time in 1939 using the *Bacillus subtilis* strain in Japan. Since 1970, *B. subtilis* and *B. licheniformis* have been extensively used for the production of alpha-amylase enzymes [1]. Amylase enzymes are significant commercial enzymes representing about 25–33% of the global enzyme market, used across many sectors [2]. In industrial applications, immobilized enzymes with thermostability are generally employed. The application areas of alpha-amylase enzymes are broad and varied. These enzymes are used in the liquefaction of starch, in baking, in the textile, paper, and fruit juice industries, and in alcohol fermentation [3].

According to data from the European Food Safety Authority (EFSA), it has been concluded that the use of the alpha-amylase enzyme, which has a wide range of applications, does not raise safety concerns under conditions appropriate for its intended use [4]. However, definitive information on the excessive and continuous consumption of the alpha-amylase enzyme is limited. In the use of enzyme supplements, it is important to carefully evaluate factors such as dosage and duration.

The *D. melanogaster* model organism, commonly known as the fruit fly, is one of the most preferred organisms for observing the effects of substances frequently introduced into our bodies from our environment. The primary reason for its preference is that it is easy and cost-effective to cultivate in the laboratory. Additionally, it is extensively used in the study of environmental pollution and toxicological responses [5]. It has been a long-studied organism for developmental biologists and geneticists due to its possession of genes homologous to humans. Particularly in toxicology studies, it has a broad application for detailing the molecular genetic mechanisms of various chemicals [6]. For example, the potential toxic effects of various substances on live and environmental health, such as thymol and carvacrol [7], various nanoparticles [8], curcumin-loaded nanocapsules [9], and heavy metals like lead [10], can be evaluated through various tests by adding these substances to the *D. melanogaster* feeding medium. One of these tests is related to developmental parameters, namely larval toxicity analysis. In this analysis, the transition of the *D. melanogaster* larva to the pupal stage and from the pupal stage to the adult fly stage, depending on the administered dose amount, is evaluated with numerical data in comparison to the control group [11,12]. One of the most preferred analyses in genotoxicity experiments is the comet assay. This test is a rapid and sensitive method used to detect DNA damage in single cells. Comet assay using *D. melanogaster* is conducted to detect DNA damage caused by various chemicals and agents [13]. Furthermore, studies conducted on *D. melanogaster* facilitate the revelation of various substances’ toxicity, cytotoxicity, genotoxicity, and teratogenic effects. Reactive oxygen species (ROS) produced in cells due to various substances affecting live cells have been associated with toxicity. High levels of ROS can damage cells in various ways. Cells possess a set of antioxidant enzymes, including peroxidase, glutathione peroxidase, and catalase, to protect against excessive ROS. If antioxidant enzymes cannot neutralize elevated ROS levels, DNA damage, cell damage, cytotoxicity, apoptosis, and uncontrolled cell regulations occur, leading to abnormal physiological and genotoxic conditions [14]. The combined application of such tests also holds significant importance in toxicity evaluations.

In this study, the effects of domestically produced alpha-amylase enzyme in Turkey on the reproductive performance, oxidative stress relationship, and DNA damage in *D. melanogaster* were revealed.

In our previous work, we identified a significant difference (*p* < 0.01) in the color characteristics and dimensions of bread between thermostable alpha-amylase produced in the *Aspergillus niger* G 2-1 isolate and commercially produced alpha-amylase and determined that the addition of five ppm of alpha-amylase showed optimum bread characteristics for dough processing [15]. However, safety assurance measures must be taken before any food-grade product is commercialized. Therefore, toxicological testing of production raw materials is necessary. Particularly, assessing the genotoxicity of additives and raw materials produced through microbial means is essential for the safety evaluation of the product [16]. This study revealed the effects of domestically produced alpha-amylase enzyme in Turkey on reproductive performance, oxidative stress relationship, and DNA damage in *D. melanogaster*.

The significance of this study lies in assessing the biosafety risks of thermostable alpha-amylase enzymes. Although microbial-derived alpha-amylases are widely used in industry, their effects on living organisms have not been adequately investigated. *D. melanogaster* is an ideal model for toxicological studies due to its short life cycle and genetic similarity.

This research aims to evaluate the developmental, genotoxic, and oxidative stress effects of alpha-amylase obtained from the *Aspergillus niger* G2-1 isolate on *D. melanogaster*. The practical rationale for conducting this study is that, despite the widespread use of this enzyme, its biosafety has not been sufficiently examined. The data obtained will provide valuable insights to enhance the safety of biotechnological applications.

## 2. Material and Method

### 2.1. Fungal Alpha-Amylase Enzyme Source

The native fungal thermostable alpha-amylase enzyme used in genotoxicity studies, derived from the *Aspergillus niger* G2-1 isolate isolated from a thermal spring source, was provided by Assoc. Prof. Dr. Arzu Taşpınar Ünal from the Department of Agricultural Biotechnology, Faculty of Agriculture, Iğdır University (Figure 1). The commercial preparation of this enzyme was produced for the first time on a pilot scale as a domestic and national enzyme within the scope of the project supported by the Ministry of Agriculture and Forestry of the Republic of Turkey, TAGEM/HSGYAD/16/A05/P01/103 [15].

### 2.2. Fungus Isolation

For pre-enrichment, diluted soil samples and water samples were incubated in a water bath at 40–45 °C for 3 h. Dilutions of the soil samples at 10^−3^, 10^−4^, 10^−5^, and 10^−6^ were placed into Petri dishes, each receiving 1 mL, followed by the addition of prepared Starch Yeast Extract Agar (SYE) (starch: 5.0 g/L, yeast extract: 2.0 g/L, KH_2_PO_4_: 1.0 g/L, MgSO_4_·7H_2_O: 0.5 g/L, and agar: 15 g/L). After incubation at 40 °C for one week, the resulting molds were counted.

### 2.3. Detection of Amylase Enzyme Presence in Fungi

Iodine serves as an indicator for amylase, with the presence of the amylase enzyme in the medium being indicated by zones forming around colonies. Therefore, after incubation, colonies that form transparent zones when iodine solution (iodine: 22 mg, potassium iodide: 80 mg, glacial acetic acid: 5 mL, distilled water: 500 mL) is applied are evaluated as amylase positive (+) [17].

### 2.4. Culture Medium and Enzyme Activity Studies

In efforts to identify a more efficient culture medium in terms of amylase activity, studies have been conducted using three different starch-containing culture media: Mycological Liquid Medium (MfLM), Stock Basal Medium (SBM), and Starch Yeast Extract Liquid Medium (SYELM).

The culture media recommended by [18,19]. Stock Basal Medium (S.B.M.) and Stock Mineral Medium (S.M.M.) were used. The SBM was prepared with the following composition per 100 mL: KH_2_PO_4_, 0.02 g; MgSO_4_·7H_2_O, 0.05 g; NH_4_H_2_PO_4_, 0.05 g; yeast extract, 0.001 g; and starch, 1 g. The SMM composition included ZnSO_4_·7H_2_O of 0.14 g/100 mL and FeSO_4_ of 0.10 g/100 mL, which was added to the SBM at a concentration of 0.1%. The prepared culture media were adjusted to a pH of 6, and the sterilization of the SBM was carried out in an autoclave at 110 °C for 25 min. For the sterilization of the SMM, the millipore filtration technique was used.

For pre-enrichment, diluted soil samples and water samples were incubated in a water bath at 40–45 °C for 3 h. Dilutions of the soil samples at 10^−3^, 10^−4^, 10^−5^, and 10^−6^ were placed into Petri dishes, each receiving 1 mL, followed by the addition of prepared Starch Yeast Extract Agar (SYE) (starch: 5.0 g/L, yeast extract: 2.0 g/L, KH_2_PO_4_: 1.0 g/L, MgSO_4_·7H_2_O: 0.5 g/L, and agar: 15 g/L). The dishes were then incubated at 40 °C for one week. After incubation, the resulting molds were counted.

For other studies aimed at determining a more efficient culture medium in terms of amylase activity, in addition to SBM, the Mycological Liquid Medium recommended by [20] was used. The components of the Mycological Liquid Medium were prepared as Bacto-soytone of 1 g/100 mL and Bacto-dextrose of 4 g/100 mL. The pH of this culture medium was adjusted to 6 and sterilized in an autoclave at 121 °C for 15 min.

A high yield and activity were observed in the modified Starch Yeast Extract Liquid Medium (peptone: 0.5 g/100 mL; beef extract: 0.15 g/100 mL; yeast extract: 0.15 g/100 mL; NaCl: 0.5 g; starch: 1 g/100 mL). The medium was sterilized in an autoclave at 121 °C for 15 min with a pH of 7 [17] and used in microorganism culturation and enzyme production studies.

### 2.5. Enzyme Production and Activity Measurements

Amylase activity is defined as the amount of enzyme that releases 1 μmol of reducing sugar (glucose) per minute under standard assay conditions, as a unit of amylase that hydrolyzes raw starch [21].

### 2.6. Stock Culture of D. melanogaster

In our study, the wild-type Oregon-R strain of *D. melanogaster* was used. *D. melanogaster* cultures were maintained in laboratory conditions at 60–70% humidity, a constant temperature of 24 °C, and a 12 h light/12 h dark cycle in a cooled incubator within culture containers. The culture medium for *D. melanogaster* development and experimental setup was prepared according to the methods described by [22,23,24]. The *D. melanogaster* experimental groups used in all analyses were as follows: Group 1—control, Group 2—alpha-amylase enzyme (25 mg/mL), Group 3—alpha-amylase enzyme (50 mg/mL), and Group 4—alpha-amylase enzyme (100 mg/mL). These concentrations of alpha-amylase enzyme were added to the culture medium to ensure oral intake by *D. melanogaster.*

### 2.7. Larval Toxicity Analysis

In all experimental groups, 3rd instar larvae of *D. melanogaster* were used for larval toxicity analysis. Larvae (72 ± 4 h old) were obtained by crossing male and female individuals in a culture medium containing only the standard food source. A total of 20 3rd instar larvae were placed in each experimental group, and the number of individuals that were able to perform the transition from the larval to the pupal stage and from the pupal to the adult stage was determined on the 15th and 30th days chronically. The experiment was repeated three times [25].

### 2.8. Biochemical Analyses

For biochemical analyses, 50 adult flies were added to each experimental group. After 15 days, 50 adult flies exposed to different concentrations of alpha-amylase in the food medium were collected. Tissue homogenization was performed on all collected flies, followed by measurements of malondialdehyde (MDA) and glutathione (GSH) levels, superoxide dismutase (SOD) and catalase (CAT) activities, and protein measurements. The experiment and analyses were conducted in three replicates.

Tissue homogenization was performed according to the method described by [26]. The prepared supernatants were used for measuring MDA, GSH, SOD, CAT activities, and protein levels. MDA levels, a marker of lipid peroxidation (LPO), were measured in tissue homogenates using the method of [27]; GSH concentration was determined according to [28]; SOD antioxidant enzyme activity was assessed using the method described by [29]; and CAT activity was measured following the method of [30]. Protein measurements were conducted using a commercially available Modified Lowry Protein Assay Kit (#SK404, Bio Basic Inc., Markham, ON, Canada). Spectrophotometric measurements were performed using the Thermo Scientific™ Multiskan™ GO Microplate Spectrophotometer (Thermo Fisher Scientific, Waltham, MA, USA).

### 2.9. DNA Damage Analysis

To detect potential DNA damage in *D. melanogaster* caused by alpha-amylase enzyme, the comet assay was performed. Fifty adult flies were assigned to each experimental group, and after 15 days of exposure to different concentrations of alpha-amylase in the food medium, samples from adult *D. melanogaster* were taken to prepare separate slides for the comet assay. This assay was repeated three times for each experiment following the method proposed by [31]. Damage levels in approximately 100 randomly selected cells per slide were evaluated on a scale from 0 (undamaged) to 4 (severely damaged), according to [32].

### 2.10. Statistical Analysis

The results obtained were analyzed with larval toxicity data reported in numerical and percentage (%) forms. The analysis of biochemical results was performed using GraphPad Prism 8 (GraphPad Software, San Diego, CA, USA), employing One Way ANOVA and Tukey’s multiple comparisons test to identify differences between groups. For the analysis of DNA damage results, Dunnett’s multiple comparisons test was used. A value of *p* < 0.05 was considered statistically significant.

## 3. Results

The transformation of *D. melanogaster* larvae into pupae and the percentages of adult fly formation from pupae were evaluated with the application of alpha-amylase enzyme (25 mg/mL, 50 mg/mL, and 100 mg/mL). When the control group was considered as 100%, it was observed that both the percentage of adult fly formation and pupa development percentage varied depending on the increase in alpha-amylase enzyme concentration (Table 1 and Table 2). In another study, it was also observed that silver and sulfur nanoparticles altered the percentages of transformation from *D. melanogaster* larvae to pupae and from pupae to adult flies [33]. In our study, it was revealed that the closest percentage of pupa and adult fly formation to the control group was at the alpha-amylase enzyme concentration dose of 25 mg/mL after periods of 15 and 30 days.

According to the biochemical analysis results, the effects of alpha-amylase enzyme exposure (at doses of 25 mg/mL, 50 mg/mL, and 100 mg/mL) for 15 days on oxidative stress parameters in adult *D. melanogaster* were examined. Looking at the values of malondialdehyde (MDA), glutathione (GSH), superoxide dismutase (SOD), and catalase (CAT) presented in Table 3, it was determined that there was no statistically significant change in these values with the increase in alpha-amylase enzyme dose when evaluated against the control group that received no substance application (*p* > 0.05).

In our study, DNA damage in adult *D. melanogaster* exposed to alpha-amylase enzyme (at doses of 25 mg/mL, 50 mg/mL, and 100 mg/mL) for 15 days was examined. Figure 2 shows the comet gel image (A) and the percentage of cell ratio (B). It was observed that there was no DNA damage in the control group and the group treated with 25 mg/mL alpha-amylase enzyme. In groups treated with 50 mg/mL of the alpha-amylase enzyme, first-class damage was observed in 20% of the cells, while in groups treated with 100 mg/mL of the enzyme, 30% of the cells exhibited first-class and 5% exhibited second-class DNA damage. According to the conditions and data of the study conducted by [34], alpha-amylase enzyme (at doses of 25–5000 μg/plate) was not mutagenic according to the Ames test result, and it did not cause chromosomal aberrations according to the in vitro chromosome aberration test result (at doses of 1000, 2500, and 5000 μg/mL of alpha-amylase enzyme).

In our study, larval toxicity data were evaluated in numerical and percentage (%) form. Statistical analyses of biochemical and DNA damage results did not reveal any significant differences between groups (*p* > 0.05). Therefore, additional representation of statistical significance levels in tables and figures was not included.

## 4. Discussion

Enzymes used in nearly every sector of industry are typically derived from microorganisms. The reasons for the widespread use of microorganism-derived enzymes include their high catalytic activities compared to plant or animal-sourced enzymes, their ability to produce without unwanted by-products, their stability and cost-effectiveness, and the possibility of obtaining them in large quantities [15].

With the continuous development of enzyme technology, the diversity of applications for products, and their high economic value, the importance of research in the field of enzymes within biotechnology is increasing. The fact that most countries are entirely dependent on imports for enzymes further emphasizes the significance of this issue [15]. Nowadays, amylases are frequently used in various industrial fields such as food, detergent, textile, paper, and pharmaceutical industries. The use of amylase enzymes in the food industry, especially in the bakery sector, is quite widespread. It is an enzyme used for many purposes, including extending the shelf life of bread, regulating the fermentation process, improving bread quality, homogenizing the bread’s internal structure, and increasing the dough’s energy value and workability [15].

Ref. [35] obtained a new strain producing alpha-amylase and reported that its use in bread-making reduced the hardness of the bread and improved the texture parameters by increasing stickiness and elasticity values [35].

Studies have determined that bread made using native alpha-amylase enzyme obtained from indigenous thermophilic fungal isolates yielded higher volume and specific volume values compared to bread to which commercial alpha-amylase enzyme was added. The differences in volume and color between bread with alpha-amylase enzyme and the control bread were found to be statistically significant (*p* < 0.01). An optimum effect in terms of dough processing was observed with the addition of 5 ppm (mg/L) alpha-amylase [15].

Many industrial microbial enzymes produced today are used in the food industry. Hazards are well defined and include bacterial, plant, and animal toxins, allergens, and antinutrients. The enzyme industry has conducted numerous genotoxicity studies. Specifically, over 230 mutagenicity studies on bacterial cells and more than 240 on mammalian cells have been performed. The requirements for the safety assessment of food enzymes are complex due to differences in global regulations. This includes variations in the conditions where pre-market approval is necessary and sensitivities to perceived risks associated with the use of biotechnology in the development of new enzyme products, leading to inconsistencies in the type of safety data that producers need to provide [36]. When enzymes consumed as food enzymes are evaluated, tests for allergenicity, genetic, and oral toxicity have not indicated any concerns regarding their use, and it has been established by the European Food Safety Authority (EFSA) that these enzymes do not create genotoxic concerns, demonstrating their suitability for use in the baking industry [37]. Ref. [38] evaluated 19 enzymes used in the food industry (such as alpha-amylases, proteases, decarboxylases, glucoamylases, lipases, glucose oxidases, pectin esterases, beta-glucanases, laccases, xylanases, and pectin lyases) for allergenicity and determined that none of the 19 enzymes were food allergens. Furthermore, enzymes such as amylase, lipase, cellulase, invertase, papain, pepsin, bromelain, lactase, superoxide dismutase, and pancreatin have been widely used as over-the-counter aids for digestion since the late 19th century [39]. There are few reported side effects and adverse events associated with these digestive enzyme supplements. However, as digestive enzymes are relatively safe, serious issues have not been encountered mostly when taken in the amounts listed on the packages [40,41]. According to a study by [42], the application of alpha-amylase enzyme in rats did not produce toxicity according to weight change, mortality rate, and limited biochemical and histopathological analysis results. Both biotechnological advancements in enzyme applications and molecular seed differentiation aim to improve agricultural efficiency, sustainability, and food security. While alpha-amylase research enhances food processing, molecular seed selection optimizes crop production, making biotechnology a crucial tool in modern agriculture.

The genotoxic effect of fungal α-amylase enzyme on *Drosophila melanogaster* may be through different mechanisms. These mechanisms include the following: (i) Fungal α-amylase can increase the production of reactive oxygen species (ROS) in cells. Increased ROS levels can cause oxidative damage to DNA, causing base mutations, double-strand breaks, and chromosomal abnormalities. (ii) Fungal α-amylase can disrupt cellular energy balance by altering sugar metabolism in Drosophila cells. Metabolic imbalances can increase error rates during DNA replication or trigger cellular stress responses, causing mutations. (iii) Enzymatic activity can affect cellular epigenetic regulations (e.g., DNA methylation and histone modifications), causing abnormal changes in gene expression. Such epigenetic changes may disrupt the division and repair processes of cells and lead to genetic abnormalities. (iv) Fungal α-amylase itself or its metabolites may directly interact with DNA and cause structural changes. (v) If it affects the gut microbiota of *Drosophila melanogaster*, this may indirectly lead to DNA damage. Disturbances in the microbiota balance may lead to the production of toxic metabolites and the triggering of inflammatory responses. This may increase cell stress and lead to changes in genetic material. These assumptions aim to explain the genotoxic potential of fungal α-amylase on *Drosophila melanogaster* and require experimental verification. Methods such as chromosome aberration analysis, comet assay, measurement of ROS levels, and gene expression analyses can be used to test these hypotheses experimentally.

## 5. Conclusions

Within the scope of this study, toxicity tests for domestically produced alpha-amylase enzyme, manufactured as a powder preparation on both laboratory and pilot scales, were conducted. When the comet assay, larval toxicity analysis, and biochemical parameters were evaluated, it was concluded that the 25 mg/mL dose of alpha-amylase had no harmful effect on the *D. melanogaster* model organism.

We consider the genotoxicity analyses conducted during our study not only as a means to determine enzyme dosing in the baking and bread-making industry but also as a pilot study that will shed light on future scientific and industrial research in the literature.

The potential genotoxic effect of fungal alpha-amylase on *Drosophila melanogaster* may be linked to oxidative stress and DNA damage mechanisms. The breakdown of starch by the enzyme releases free sugars, which can enhance cellular metabolism and lead to the accumulation of reactive oxygen species (ROS). Excessive ROS production can cause lipid, protein, and DNA damage, thereby triggering genotoxicity.

Additionally, enzymatic by-products may disrupt cellular homeostasis, placing stress on antioxidant defense mechanisms and potentially affecting DNA repair systems. Some metabolites derived from *Aspergillus niger* could also contribute to DNA damage, even at low concentrations.

In conclusion, while low-dose alpha-amylase appears to be non-toxic, higher concentrations may induce DNA damage through oxidative stress. Further research is needed to elucidate this mechanism by investigating oxidative stress markers and DNA repair gene expression.

## Figures and Tables

**Figure 1 biology-14-00219-f001:**
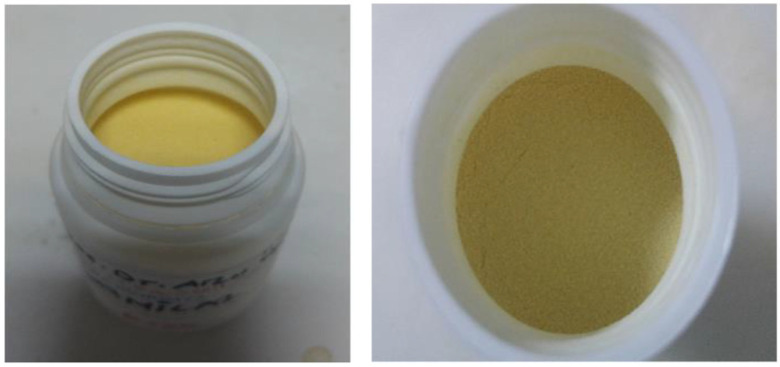
Powdered form of alpha-amylase obtained from the *Aspergillus niger* G2-1 isolate (Ünal et al., 2022 [15]) (ORBA Biochemistry Industry and Trade Inc., Istanbul, Türkiye).

**Figure 2 biology-14-00219-f002:**
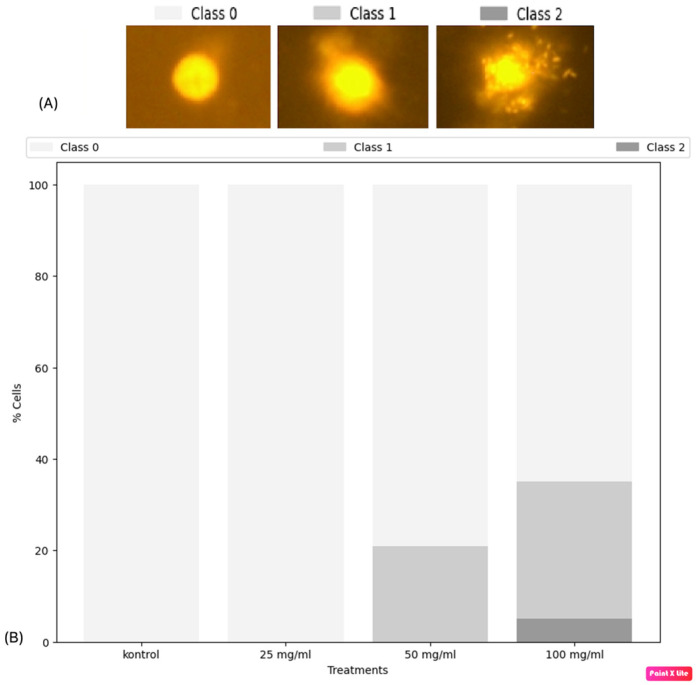
The effect of different concentrations of alpha-amylase enzyme on DNA integrity in adult *D. melanogaster* after 15 days of exposure: (**A**) comet assay image, (**B**) DNA damage rate.

**Table 1 biology-14-00219-t001:** The effect of alpha-amylase enzyme on the survival rate of *D. melanogaster*’s larvae (Group 1: control, Group 2: 25 mg/mL, Group 3: 50 mg/mL, Group 4: 100 mg/mL) (15 days).

Experiment Groups	Adult *D. melanogaster*						Pupa	
	Female	%	Male	%	Total	%	Growty	%
**Group 1**	14	100	13.6667	100	27.6667	100	37	100
**Group 2**	14.3333	102.381	13	95.122	27.3333	98.7952	44.6667	120.721
**Group 3**	11	78.5714	11	80.4878	22	79.5181	33	89.1892
**Group 4**	9.33333	66.6667	6.66667	48.7805	16	57.8313	27.6667	74.7748

**Table 2 biology-14-00219-t002:** The effect of alpha-amylase enzyme on the survival rate of *D. melanogaster*’s larvae (Group 1: control, Group 2: 25 mg/mL, Group 3: 50 mg/mL, Group 4: 100 mg/mL) (30 days).

Experiment Groups	Adult *D. melanogaster*						Pupa	
	Female	%	Male	%	Total	%	Growty	%
**Group 1**	34.6667	100	44.3333	100	79	100	127	100
**Group 2**	41.3333	119.231	38.3333	86.4662	79.6667	100.844	115	90.5512
**Group 3**	33	95.1923	35	78.9474	68	86.0759	107.333	84.5144
**Group 4**	13	37.5	19	42.8571	32	40.5063	75.3333	59.3176

**Table 3 biology-14-00219-t003:** Effects of alpha-amylase enzyme on malondialdehyde (MDA), glutathione (GSH), superoxide dismutase (SOD), and catalase (CAT) levels in *D. melanogaster* tissues (Group 1: control, Group 2: 25 mg/mL, Group 3: 50 mg/mL, Group 4: 100 mg/mL).

Experiment Groups	MDA (nmol/g Tissue)	GSH (nmol/g Tissue)	SOD (U/µg Protein)	CAT (nmol min^−1^/µg Protein)
**Group 1**	1.67 ± 0.02	14.60 ± 1.41	1.42 ± 0.11	1.71 ± 0.37
**Group 2**	1.75 ± 0.09	14.43 ± 0.87	1.54 ± 0.15	1.91 ± 0.35
**Group 3**	1.81 ± 0.22	14.26 ± 1.37	1.57 ± 0.18	1.95 ± 0.53
**Group 4**	1.86 ± 0.14	13.85 ± 1.60	1.58 ± 0.13	2.19 ± 0.82

## Data Availability

The original contributions presented in this study are included in the article. Further inquiries can be directed to the corresponding author.

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
