# Peer review of "Research on Genotoxicity Evaluation of the Fungal Alpha-Amylase Enzyme on Drosophila melanogaster"

_biology, 2025, doi:10.3390/biology14030219_

Round 1
Reviewer 1 Report
Comments and Suggestions for Authors
The study by Arzu Ünal et al. investigates the genotoxic effects of a thermostable alpha-amylase enzyme from Aspergillus niger on the model organism Drosophila melanogaster. The research aims to evaluate the effects of the enzyme on DNA integrity, developmental process and oxidative stress. Despite the wide range of experimental methods, the manuscript lacks justification for the practical and scientific significance of the study. The focus on a single enzyme limits the generalizability of the findings to other enzymes or broader contexts within biotechnology. The study could benefit from a more detailed discussion of the potential mechanisms of the observed genotoxicity.
Comments:
Please indicate in the introduction what is the relevance of the study and what is the practical reason for conducting it?
Please state the scientific significance of the study and what new knowledge it provides.
Please state the hypothesis of the study.
In the results section, the tables and figures should indicate the levels of statistical significance of the observed differences, where applicable.
In the discussion section, please consider what is the putative mechanism of genotoxic action of fungal alpha-amylase enzyme on Drosophila melanogaster.
Comments on the Quality of English LanguagePlease translate the header of tables 1 and 2.
Author Response
Dear Reviewer,
All corrections (red colour) have been made as requested.
The article is attached.
Kind regards,
Arzu ÜNAL

Reviewer 2 Report
Comments and Suggestions for Authors
The manuscript by Ünal et al devoted to the investigation of a potential toxicity of the domestically produced alpha-amylase enzyme in Turkey. The authors used Drosophila melanogaster as a model object to perform their study and estimated its reproductive performance, oxidative stress relationship and DNA damage. The results obtained demonstrated that feeding with higher concentration of alpha-amylase (50 mg/ml and 100 mg/ml) resulted in a severe decrease in the survival rate of D. melanogaster’s larvae and an increased of DNA damage rate in imagoes. However, the oxidative stress parameters in adult D. melanogaster were not altered after the same alpha-amylase’s treatment.
The study is well designed, appears to be well executed and contains new data that would be of interest to the readers of Biology.
However, I would like to draw attention to one important point. The authors stated in the Abstract that the low dose of alpha-amylase (25 mg/ml) did not cause genotoxicity, oxidative stress and decrease of survival, but did not mention that higher doses did demonstrate negative effects. This fact should also be reflected in the Abstract.
Minor comments
Throughout the text: “D.melanogaster” instead of “D. melanogaster”.
Lines 168-170:
the verb is missing – “…the number of individuals that could transition from the larval to the pupal stage and from the pupal to the adult stage was recorded on the 15th and 30th days chronically”
should be “are able to perform transition” or “are able to transit”
Tables 1 and 2: I don’t understand what “GeliÅŸim” means.
Comments on the Quality of English LanguageI believe that the text needs editing in English because in its current form it is hard to read and understand.
Author Response

(The authors gave the same response as above.)

Reviewer 3 Report
Comments and Suggestions for Authors
The manuscript "A Research on Genotoxicity Evaluation of the Fungal Alpha-2 Amylase Enzyme on Drosophila melanogaster" presents interesting results. The Manuscript was prepared carefully considering the new and distinctive number of literature positions on the subject of interest. In my opinion, the text was written properly and information was collected in the appropriate manner. Please note that some corrections have been made as examples, but it is the authors' responsibility to ensure that all similar corrections are addressed in the Manuscript.
However, before being considered for publication, the work needs some improvement. Some typos mistakes are present in the manuscript, for instance, and not exhaustively:
The manuscript is generally well-written in English. However, there are some flaws that may need to be addressed. I kindly requested the authors' assurance that a native English speaker would be engaged to review the manuscript, as it required moderate English editing.
I believe this paper can be accepted after a revision.

The manuscript is generally well-written in English. However, there are some flaws that may need to be addressed. I kindly requested the authors' assurance that a native English speaker would be engaged to review the manuscript, as it required moderate English editing.
Author Response

(The authors gave the same response as above.)

Round 2
Reviewer 1 Report
Comments and Suggestions for Authors
Dear Authors,
Overall, the comments have been addressed and the manuscript has been improved. However, some minor comments remain.
The order of paragraphs in the introduction should be changed. The general part should come first, then the hypothesis and objectives, and the results at the end.
The added text in the discussion should include references to sources of literature.
Author Response
Dear Reviewer,
The final version of our manuscript, with the requested revisions made, is attached.
Best regards,
Arzu TAÅžPINAR ÜNAL
